# LSTM-Based DWBA Prediction for Tactile Applications in Optical Access Network

**Elaiyasuriyan Ganesan** [1], **Andrew Tanny Liem** [2], **I-Shyan Hwang** [1,*], **Mohammad Syuhaimi Ab-Rahman** [3], **Semmy Wellem Taju** [2] and **Mohammad Nowsin Amin Sheikh** [1]

1.  Department of Computer Science and Engineering, Yuan Ze University, Taoyuan 32003, Taiwan
2.  Department of Computer Science, Universitas Klabat, Manado 95371, Indonesia
3.  Department of Electrical and Electronic Systems Engineering, Faculty of Engineering and Build Environment, Universiti Kebangsaan Malaysia, Bangi 43600, Selangor, Malaysia
*   Correspondence: ishwang@saturn.yzu.edu.tw

**Abstract:** Historically, the optical access network (OAN) plays a crucial role of supporting emerging new services such as 4 k, 8 k multimedia streaming, telesurgery, augmented reality (AR), and virtual reality (VR) applications in the context of Tactile Internet (TI). In order to prevent losing connectivity to the current mobile network and Tactile Internet, the OAN must expand capacity and improve the quality of Services (QoS) mainly for the low latency of 1 ms. The optical network has adopted artificial intelligence (AI) technology, such as deep learning (DL), in order to classify and predict complex data. This trend mainly focuses on bandwidth prediction. The software-defined network (SDN) and cloud technologies provide all the essential capabilities for deploying deep learning to enhance the performance of next-generation ethernet passive optical networks (NG-EPONs). Therefore, in this paper, we propose a deep learning long-short-term-memory model-based predictive dynamic wavelength bandwidth allocation (DWBA) mechanism, termed LSTM-DWBA in NG-EPON. Future bandwidth for the end-user is predicted based on NG-EPON MPCP control messages exchanged between the OLT and ONUs and cycle times. This proposed LSTM-DWBA addresses the uplink control message overhead and QoS bottleneck of such networks. Finally, the extensive simulation results show the packet delay, jitter, packet drop, and utilization.

**Keywords:** OAN; Tactile Internet (TI); deep learning (DL); SDN; LSTM-DWBA; QoS

## 1. Introduction

The transmission medium of the optical access network is optical fiber, the backbone of today's high-speed Internet with the replacement of the traditional copper wires. Globally deployed optical access networks (OANs) meet the higher capacity, reliable, and more secure transmission requirements of high-bandwidth demand video streaming and cloud services in past decades. Mainly 10 Gbps-based fiber-to-the-home (FTTH) has become a popular service in many cities in many countries, particularly the countries such as South Korea, Japan, and China [1]. Further, the OAN plays a crucial role in the fiber-to-the-X (FTTx) and industrial networks.

However, the recent emergence of Tactile Internet (TI) applications, such as telesurgery, high-definition 8K UHD video streaming, autonomous vehicles, virtual and augmented reality, online gaming, and many others present significant challenges to OANs when it comes to providing services with an assured network resource provision, ultra-low latency [2]. According to the Cisco virtual network forecast, by 2022, Internet video traffic will account for 82% of all consumer Internet traffic, up from 73% in 2017 [3]. The proliferation of new services, the unimaginable growth in Internet video traffic, and the rapid development of the backbone network have all contributed to an intensification of the bottleneck in the first and last mile of Internet connectivity. The next-generation optical

access network developed by the IEEE 802.3ca 25 G and 50 G NG-EPON is a promising solution for the above-presented problems [4]. Furthermore, NG-EPON is regarded as the most attractive solution for removing the constraints of the optical network and providing higher bandwidth, low latency, and support for stringent QoS requirements for a wide variety of new and emerging applications (e.g., Tactile Internet and Internet of Things) [5].

NG-EPONs newly introduced a two wavelength channel to provide a higher bandwidth capacity. It allows the operation of Ethernet passive optical networks (EPONs) to be expanded to multiple channels of 25 Gb/s, enabling the transmission of data at the following data rates downstream and upstream): 25/10 Gb/s, 25/25 Gb/s, 50/10 Gb/s, 50/25 Gb/s, 50/50 Gb/s, and 50/50 Gb/s. Furthermore, the channel bonding technique is introduced to enable optical network units (ONUs) to combine multiple wavelength channels for increased system throughput [6]. In addition, introduces a multi-channel reconciliation sublayer (MCRS) that allows multiple media access controls (MACs) to interact with multiple physical layers. The MCRS is the multipoint MAC control (MPMC) sublayer for such PONs, which is responsible for the management, user registration, and mainly for bandwidth allocation. The MPMC consists of the following two main protocols: (1) multipoint control protocol (MPCP), responsible for arbitration of TDM-based access to the point-to-multipoint (P2PM) medium; (2); channel control protocol (CCP), which is responsible for querying and controlling multiple channels within Nx25G-EPON PHY [7]. IEEE classifies hybrid-EPON as a multi-scheduling domain (MSD), single scheduling domain (SSD), or wavelength agile (WA)-EPON for effective scheduling and bandwidth utilization in next-generation EPON. It is dependent upon how wavelength and bandwidth are allocated/managed by multiple channels in the single ONU or group of ONUs in the dynamic wavelength bandwidth allocation (DWBA) mechanism. The MPCP is responsible for timing and arbitrating the ONU transmissions. The second protocol is channel control protocol (CCP), which is responsible for querying and controlling multiple channels within Nx25G-EPON PHY. It is also more reliable and efficient than the old EPON system [6].

The main contribution of this paper is as follows:

1. We proposed the SDN-enhanced NG-EPON architecture and operations;
2. We propose a novel DWBA scheme that employs long-short term memory-dynamic wavelength bandwidth allocation (LSTM-DWBA) for emerging Tactile Internet applications into the network;
3. We designed the LSTM-DWBA scheme as an offline scheduler with inter- and intra-traffic scheduling mechanisms;
4. We build an LSTM model and train it into the bandwidth requests for the next cycle based on the past cycle's historical data;
5. The extensive simulation results LSTM-DWBA outperform the without prediction DWBA scheme (normal DWBA) in terms the accuracy;
6. More specifically, the LSTM-DWBA scheme is reducing the bandwidth overhead and improves the bandwidth utilization;
7. Furthermore, LSTM-DWBA can gain more users and tactile services in the network

Moreover, in this paper, we only predict the bandwidth demands for the next cycle because we use traffic priority and burst traffic.

The remainder of this paper is organized in the following manner. Related work is presented 2. The overview of LSTM architecture is presented in Section 3. The proposed NG-EPON architecture is described in Section 4. Section 5 discusses the performance evaluation and simulation. Section 6 brings our work to a conclusion.

## 2. Related Work

Today's optical networks are required to provision bandwidth speedily and with accurate bandwidth prediction in order to facilitate network resource utilization and maintenance. Recently, ML techniques have been successfully used in the development of optical access networks. AI is useful for tackling the latency problem of optical networks for tactile applications due to its powerful modeling capabilities [8]. The network advancement

of the SDN and graphics processing unit (GPU) and the cloud computing technologies to used store and train the data computationally using machine learning models such as deep learning architectures [9]. For example, ML-based predictive bandwidth allocation is proposed for low-latency applications using the artificial neural network (ANN) models proposed. An ANN model is used in the OLT to detect the ON and OFF periods of Internet traffic in the next polling cycle of each ONU. Based on this prediction, the bandwidth demand of waiting time is estimated [10]. The author of [11] proposed an AI-based bandwidth allocation for human-to-machine (H2M) applications over the future access networks, where Tactile Internet real-time haptic and feedback traffic traces are collected and predicted to allocate the bandwidth. Recently, Theresal et al. proposed a GRU-recurrent neural network (RNN)-based dynamic bandwidth allocation (DBA) scheme for XG-PON as a pioneer in C-RAN network, which provides a better result in terms of latency, packet loss, and jitter [12]. The GRU model is trained to predict the uplink latency in the mobile fronthaul C-RAN network. Furthermore, we studied the ML/DL-based PON DBA mechanism in [13–16].

The majority of the existing DL models for network traffic classification are based on specific neural network architectures, namely, and recurrent neural networks (RNNs) and their specific variants as follows: the LSTM work [17], and the gated recurrent neural network (GRU) [15]. The author [18] proposes an innovative resource allocation framework for virtualized network environments. AI techniques such as convolutional and LSTM networks to allocate resources to improve performance and reduce cost. Some other works classified the CNN/LSTM-based network traffic classification on IoT networks in [19]. LSTMs solve the vanishing gradient problem by introducing specific core elements (i.e., gates) that allow gradients to flow unchanged through the network during training. As a result of their ability to effactully capture nonlinear long-term dependencies in data sequences, LSTMs have emerged as a promising choice for a variety of time series forecasting problems.

## 3. Overview of Long Short-Term Memory Architecture

Recurrent neural networks (RNNs) are used to recognize patterns in time series data. Given the current and previous state, the latent state of the RNN at a time step $t$ can carry forward the memory to predict the next time step ($t + 1$) [20]. Further, RNN has good performance in communication modeling of time-varying data and is suitable for processing signals of optical fiber communication systems obtained in time series format. LSTM networks are a special type of RNN capable of learning both short-term and long-term dependencies. These networks work well on a variety of problems such as addressing the vanishing and exploding gradient problems of conventional RNN [21,22].

The key state of the LSTM architecture is a set of memory blocks, as shown in Figure 1. Each block contains a cell state, and adding and removing information to or from the cell state is accomplished through gates. This consists of a pointwise multiplication operation and a layer of a sigmoid neural network. The output of the sigmoid layer is between 0 and 1, with 0 indicating no information passing through and 1 indicating all information passing through. The LSTM model starts when the long-term memory (LTM) and short-term memory ($STM$) come through in time sequence $t - 2$ i.e., $LTM_{t-2}$ and $STM_{t-2}$ and then, an event and an output are coming in and out of the LSTM, i.e., $LTM_{t-1}$ and $STM_{t-1}$, passing to the next node, and so forth, thus keeping track of the $LTM$ and $STM$. Note that the following output of $LTM_{t-1}$ and $STM_{t-1}$ is the updated output from $LTM_{t-2}$ and $STM_{t-2}$ and the prediction $Output_{t-1}$. The LSTM contains the following four gates: Learn gate, the Forget gate, the Remember gate, and the Use gate.

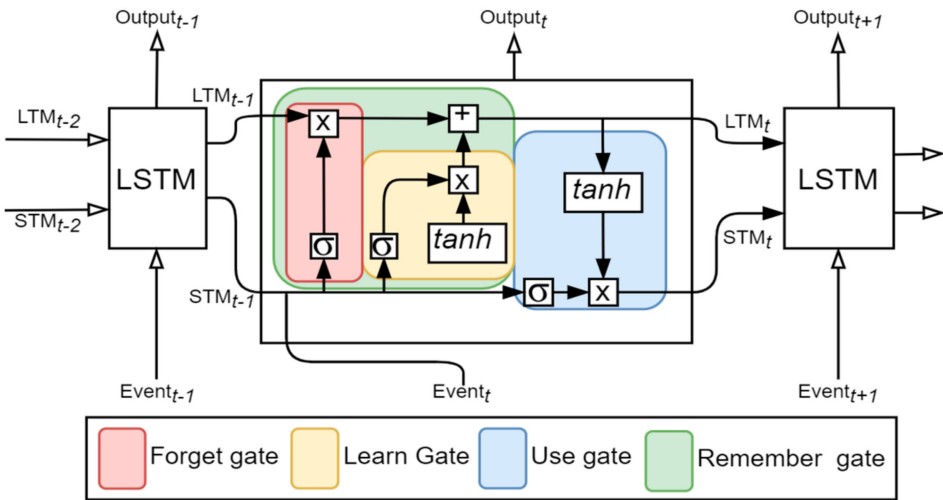

**Figure 1.** Architecture of LSTM.

The First gate is Learn gate will combine an $STM_{t-1}$ and a current event ($E$), retaining only the relevant information. The E represents the new input data to be predicted. The Learn gate's output is $N_t i_t$ where:

$$N_t = tanh(W_n[STM_{t-1}, E_t] + b_n) \tag{1}$$

$$i_t = \sigma(W_i[STM_{t-1}, E_t] + b_i) \tag{2}$$

where learn matrix $N_t$ and ignore factor $i_t$ are multiplied together to generate the Learn gate result.

Second, a forget gate is used to determine which information should be kept and which should be forgotten. It takes previous long-term memory ($LTM_{t-1}$) as input and decides what information to keep and what to forget. The information from the previous $STM_{t-1}$ and current input $E$ is passed through the sigmoid ($\sigma$) activation function. If the value is closer to 0 means to forget, and closer to 1 means to retain the information. The following Equation (3) are described as Forget gate.

$$f_t = \sigma\left(W_f[STM_{t-1}, E_t] + b_f\right) \tag{3}$$

where $f_t$ represent the forgot factor. The $f_t$ is multiplied with the previous $LTM_{t-1}$ to produce the Forget gate output.

Third, the Remember gate combines the $LTM$ from the Forget gate and the $STM$ from the Learn gate. Therefore, the output of Remember gate is as follows:

$$LTM_t = LTM_{t-1} . f_t + N_t i_t, \tag{4}$$

where $N_t$, $i_t$ and $f_t$ are calculated in Equations (1)–(3).

Finally, the Use gate to combine vital information from previous $LTM$ memory and Previous $STM$ memory to generate $STM$ for the next and cell and output for the present event. This gate will take what is helpful from the $LTM$ and $STM$ and update the $STM_t$, thus the output of the Use gate is $U_t V_t$, where

$$U_t = tanh(W_u LTM_{t-1} . f_t + b_u) \tag{5}$$

$$V_t = \sigma(STM_{t-1}, E_t + b_v) \tag{6}$$

where, is a dot product operation, $W_n$, $W_i$, $W_u$, $W_f$ are weight values between the current and previous hidden layers of Learn gate, Use gate, and Forget gate. $b_n$, $b_u$, $b_f$ are offset

vectors between the current and previous hidden layers of Learn gate, Use gate, and Forget gate. $U_t$, $V_t$ is the output of the Use gate.

## 4. Proposed System Model

In this section we presented the Tactile Internet supported SD-NG-EPON architecture and mechanisms.

### 4.1. Tactile Internet and Cloud-Based SD-NG-EPON Architecture and Operations

Recently, the demand for bandwidth has increased due to the global pandemic COVID-19 and emerging new applications. Hence, we propose an SD-NG-EPON network architecture and DWBA mechanism to meet these requirements. Figure 2 shows the proposed network architecture of Tactile Internet and cloud-based SD-NG-EPON. The architecture consists of IEEE 802.3ca-based 25G Gb/s with a typical fiber range of 20 km between the central software-defined optical line terminals (SD-OLT) and software-defined optical network units (SD-ONUs). The EPON may comprise multiple stages, each stage separated by a wavelength-broadcasting splitter/combiner or wavelength multiplexer/demultiplexer. Channel bonding is an important feature for incremental speed increases of NG-EPON, while also adding to its long lifespan. NG-EPON ONUs can work using two wavelength channels simultaneously, which is totally different from traditional EPON ONUs. However, to achieve the ultra-low end-to-end latency goal of the Tactile Internet, we consider placing the tactile control server, media server, and cache server in the central office (CO), as seen in Figure 2. For local H2M teleoperation, the distance between master and slave devices is often only a few meters; therefore, wireless technologies such as 5 G, WiFi, or Bluetooth can enable it. However, when remote H2M teleoperation scenarios are considered, such as intra-PON master-slave devices (shown by the yellow aerow) and local-PON master–slave devices (shown by the pink aerow), optical front/back-haul segments become important. In this system, we divided it into three different services, i.e., application service, connection service, and transport services [23].

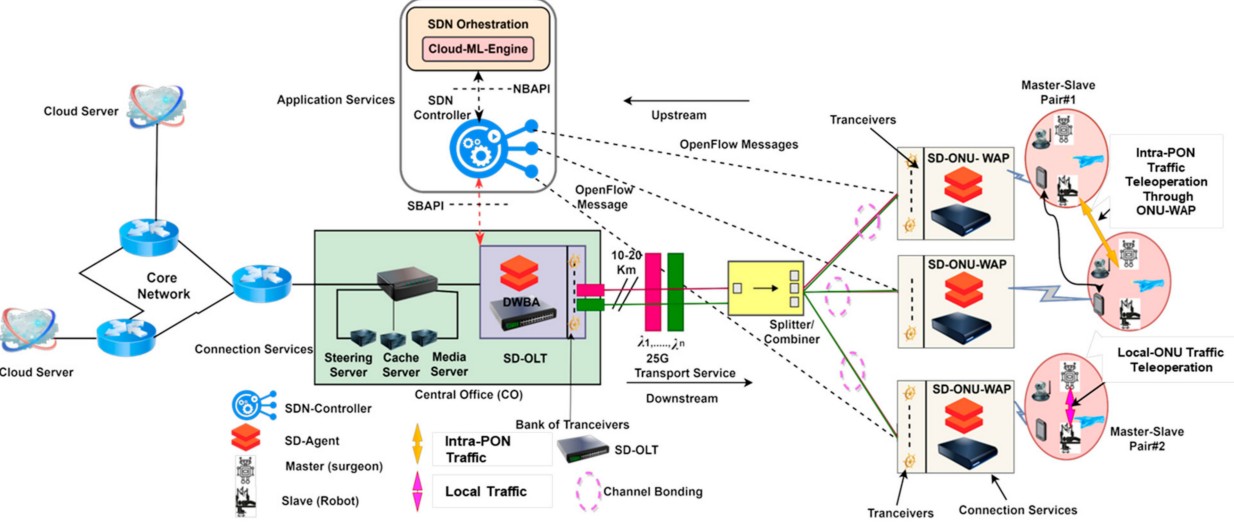

**Figure 2.** Proposed high–level architecture of cloud–based SD–NG–EPON.

#### 4.1.1. Application Services

This service is responsible for providing differentiated services to clients in order to meet their needs. It is responsible for sending and receiving control information packets from the client's applications to the SDN controller. Through northbound API, the SDN controller is able to communicate with clients (SDN management apps) (Northbound application programming interface-NB-API). Through the NB-API, the SDN management application will receive all feedback from client applications and provide it back to the

SDN controller. Southbound API allows the controller to communicate with the SD-OLT (SB-API).

The SDN controller and the SD-OLT manage the DWBA module and QoS services. The dynamic bandwidth allocation (DBA) scheme is an offline approach is utilized. The OLT may obtain all REPORT messages from the ONUs because the DBA offline method is applied. The SDN controller is used to generate a network that is intelligently managed and capable of supporting network slices for various technologies. In addition, the SDN controller will transmit this information to the cloud-based ML engine (i.e., Google Cloud, Amazon Web Services). The cloud-based ML will subsequently add this information to the LSTM network, continuing the learning process. The LSTM model is able to recognize and learn the profound numerical relationships of multistep time series, which gives it the ability to forecast future network behavior with a high degree of accuracy based on the granularity of historical network data. Finally, the DWBA module will provide efficient packets. The SD-OLT has all the traffic patterns of each SD-ONUs, stores them in the ML engine, and uses this data to improve the prediction model. All information such as traffic patterns and ML engine are communicated to the SD-OLT by the SDN controller, which is orchestrated in the SDN application [24] in the application services.

### 4.1.2. Connection Service

The SD-OLT has equipped with the two transceivers with the $\lambda_1$ and $\lambda_2$ wavelengths. The SD-ONU links subscribers to the SD-OLT through two bonded transceivers.

### 4.1.3. Transport Service

These services integrate all application and connection services into a hybrid access network. The SD-NG-EPON to adapt and support network slices across different systems, applications, and vendors. SD-ONUs have two bonded transceiver channels to provide up to 50 gb/s total transmission between SD-ONUs and SD-OLT.

### *4.2. LSTM–DWBA System Model*

In SD-NG-EPON systems, SD-OLT, and SD-ONU communicated via the multipoint control protocol (MPCP) messages such as REPORT and GATE. Each ONU sends a REPORT message to the OLT every cycle time, containing the EF, TI, AF, and BE buffering queue occupancies, which reflect end-user bandwidth demands. As a result, time slots are provided to SD-ONUs by the SD-OLT in the next cycle using GATE messages; these time slots are sized according to DWBA regulations. Figure 3 shows the proposed LSTM-DWBA ML system model. The coming in traffic from users, the ONU REPORT message, and the GATE message are essential factors in predicting the bandwidth demand for each SD-ONU [5,25]. The first is based on ONUs intra-scheduling traffic, while the second and third are used in the DBA algorithm and the network architecture and settings such as how many wavelengths will be used, the maximum cycle time, etc.

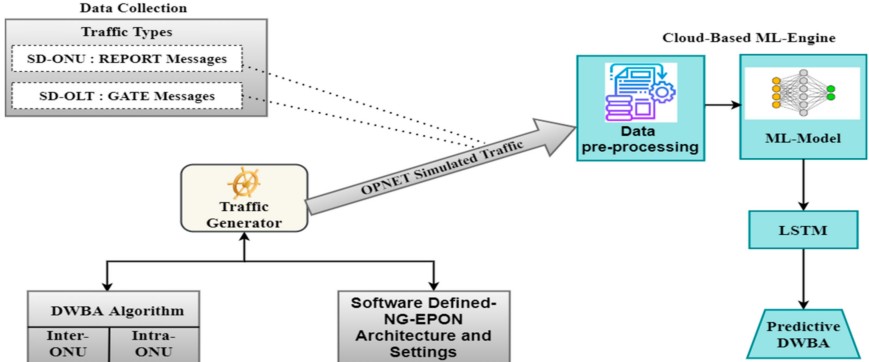

**Figure 3.** ML system model.

### 4.2.1. LSTM–DWBA Predictive Mechanism

In PON, Internet traffic can be seen as a time series that shows how much bandwidth is being used in each cycle. So, predictive DWBA can be performed with any ML model that can make predictions from one-time series to the next. The goal of using the LSTM architecture in the DWBA mechanism is to predict the future bandwidth to be allocated based on historical messages from SD-ONUs and SD-OLTs. The terms of historical messages, such as GATE and REPORT message, i.e., the long-term history of ONU(s) REPORT messages, i.e., LTM, the short-term or recent ONU(s) REPORT messages, i.e., STM and the current ONU(s) REPORT messages, i.e., E. To predict the GATE message $E_t$ for ONU(s) at time sequence t, the LSTM module will use the $LTM_{t-1}$, and the $STM_{t-1}$, so, the $LTM_{t-1}$ and $STM_{t-1}$ will give a hint or estimate of the Expedited Forwarding (EF), Tactile Internet (TI), Assured Forwarding (AF), and Best effort (BE) bandwidth. Moreover, these three pieces of information, such as the $LTM_{t-1}$, $STM_{t-1}$, and the $E_t$, will update the $LTM_t$ and the $STM_t$ module. Figure 4 shows how the LSTM architecture works as part of the LSTM-DWBA prediction mechanism.

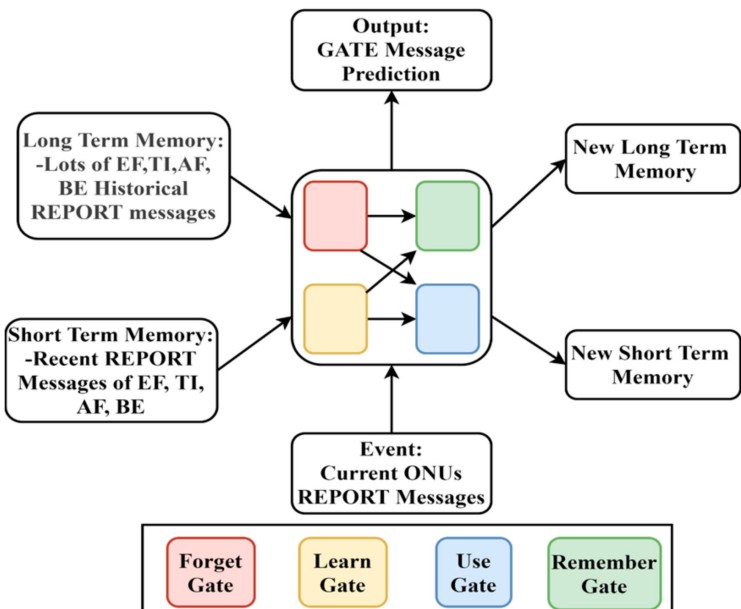

**Figure 4.** LSTM architecture for DWBA prediction mechanism.

### 4.2.2. LSTM-DWBA Operations

The proposed LSTM-DWBA operation is based on LSTM time series prediction. The main principle of Deep-DWBA is to use the predictions made by an ML model using the past P REPORT to allocate bandwidth for the next future cycles without requiring any report messages within those cycles. Thus, developing the predictive model using the LSTM in every cycle would increase the DWBA calculation time and high computational complexity. Therefore, we use the sliding learning windows based on historical observation. As a result, the proposed LSTM-DWBA has the following two cycles: the initialized cycle $i_1, i_2, i_3, \ldots, i_n$, and the prediction next cycle $i + 1$ In other words, the LSTM-DWBA can predict the GATE message multistep based on the historical observations sliding window. Therefore, idle time only exists when the LSTM prediction model is updated via the SDN controller to improve the model's accuracy. The LSTM-DWBA is based on the LSTM model, which means it is constantly learning what is new and forgetting what is not. As a result, the LSTM-DWBA improves day by day.

---

**Algorithm 1: LSTM Based-DWBA**

---

N    Number of ONUs

$I_t$         ONU report message at the *I*th cycle

$G_t$        Number of grants received in the *I*th cycle from OLT

$P_{t+1}^j$       Grant Assignment for next cycle

$\hat{E}_{t+1}^j$       Number of GRANTs (bandwidth) arrive at the ONU in the next cycle predicted by LSTM

$\hat{P}_{t+1}^j$       Pre-Grant Assignment for the *J*th

$T_r$        Tr is traffic type where $\mathrm{Tr} \in \{EF, TI, AF, BE\}$

      **Input:** Time series report message 1: $I_{t-(i+1)}$, $I_{t-(i+2)}$,      $I_{t-1}$, $I_t$

: Time series Grant message 2: $G_{t-(i+1)}$, $i_{t-(i+2)}$,     $G_{t-1}$, $G_t$

**Output:** Grant Assignment $P_{t+1}^j$ for next cycle

**I. Prediction Phase**

1. Calculate: $= + \left( I_{(t-1)} - B_{(t-1)} \right)$

2. Get time series report 3: $\{ E_{t-(i+1)}, E_{t-(i+2)}, \quad E_{t-1}, E_t \}$

3. for $(j = 1, j \leq N, J++)$

$\hat{E}_{t+1}^{j,Tr} = (f \left( E_{t-(i+1)}^{j,Tr}, E_{t-(i+2)}^{j,Tr}, \quad E_t^{j,Tr} \right)$

**4. End**

**II. Grant Assignment Phase**

5. Input: $\hat{E}_{t+1}^{j,Tr}, + I_t^{j,Tr}, G_t^{j,Tr}$

6. $\hat{P}_{t+1}^j = \hat{E}_{t+1}^{j,Tr} + \left( I_t^{j,Tr} - G_t^{j,Tr} \right)$

7. **While** $(\sum_j \hat{P}_{t+1}^{j,Tr} > Maxium\ bandwidth\ (grant))$ **do**

$\hat{P}_{t+1}^j - -$

**8. End**

9. $P_{t+1}^{j,Tr} = \hat{P}_{t+1}^{j,Tr}$

10. $\hat{P}_{t+1}^{j,Tr}, T_r \in \{EF, TI, AF, BE\}$

---

Moreover, the NG-EPON GATE and REPORT data files are stored in application services in a cloud-based ML engine day by day. With the given event time $E_t$ at the store, the data $t - 1$, and the initialize cycle i, shown in Figure 2, the cloud-based ML engine performs predictive modeling and updates the LSTM-DWBA prediction model for $d_t$ (predict and update time to the model) in the DWBA module via SDN controllers. Figure 5 depicts the timing diagram of the proposed LSTM-DWBA upstream scheduling. As shown in Algorithm 1, each DBA cycle has one prediction phase and one grant assignment phase. The $\hat{E}_{t+1}^j$ LSTM predicts the range of grants that may be awarded to the ONU during the next cycle in the prediction section. In the grant (i.e., bandwidth) assignment phase, $\hat{P}_{t+1}^j$, the pre-grant assignment was computed as the sum of $\hat{E}_{t+1}^j$ and $I_t - G_t$. If $\hat{P}_{t+1}^j$ exceeds the maximum bandwidth (MB) the value of $\hat{P}_{t+1}^j$ is reduced by one until it is accommodated by the MB. Finally, in the next cycle *i+1*, the predicted $\hat{P}_{t+1}^{j,Tr}$, bandwidth is allocated to each traffic type $T_r \in \{EF, TI, AF, BE\}$. The bandwidth allocation for SD-ONUs in standard DWBA upstream scheduling is calculated using the following formula [26,27]:

$$B_{min} = R_N \cdot \left( \frac{T_{cycle}^{max}}{N} \right) - G, \tag{7}$$

where $B_{min}$ is to guarantee bandwidth for each SD-ONU, $R_N$ denotes the transmission speed (bits/s), $T_{cycle}^{max}$ is the maximum cycle time, $N$ is the number of SD-ONUs, and $G$ is the guard time. Consequently, although the requested timeslot size from an SD-ONU exceeds the predefined $B_{min}$, the SD-OLT only grants no more than $B_{min}$. Moreover, the

SD-OLT must wait for the report message from all SD-ONUs before starting to calculate the bandwidth. Therefore, the idle time is calculated as follows [28].

$$T_{idle} = T_{DBA} + RTT + T_{ONU}, \tag{8}$$

where $T_{DBA}$ denotes the DBA calculation time and RTT denotes the round-trip time and $T_{ONU}$ is the processing of SD-ONU time (i.e., on receiving a GATE message). On the contrary, in the proposed LSTM-DWBA scheduling, after the initialization steps, the OLT directly sends the GATE message to all ONUs without waiting for the REPORT messages from SD-ONUs for the next cycle, and so on. The proposed LSTM-DWBA will grant the timeslots to each SD-ONU based on the predicted bandwidth from the LSTM model that has been calculated in the cloud-based ML engine. In this way, the upstream REPORT message overheads, and the guard time between SD-ONUs is eliminated, thus increasing the efficiency of the DWBA upstream scheduling.

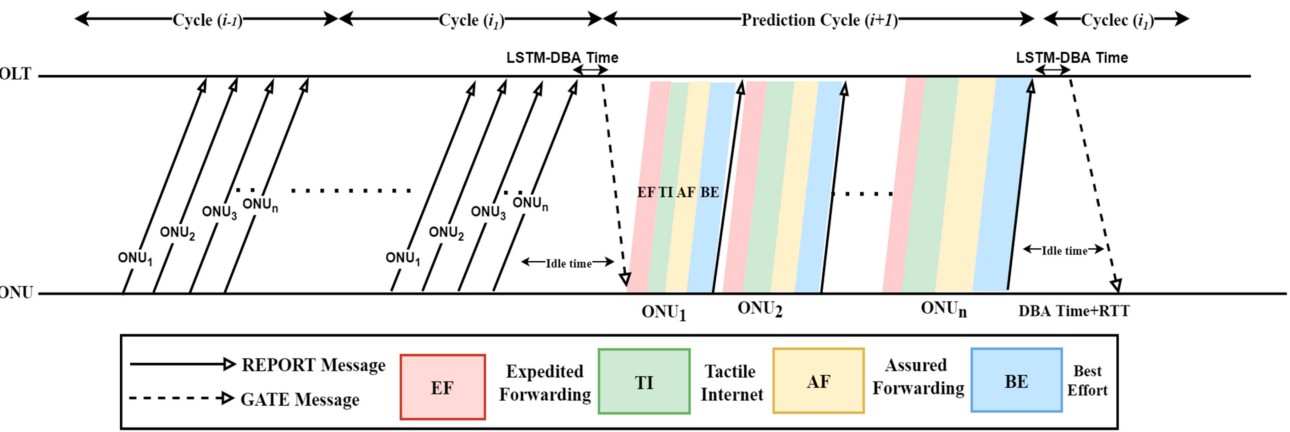

**Figure 5.** Prediction LSTM-DWBA upstream scheduling mechanism.

Therefore, the proposed LSTM-DWBA aims not only to improve QoS but also to reduce the overhead associated with the control channel in the traditional grant/report mechanism. The overhead introduced by the GATE and REPORT messages is referred to as control channel overhead. The number of scheduled SD-ONUs and the cycle time have an impact on this overhead. There are several other overhead components in NG-EPON such as burst mode overhead, forward error correction (FEC) encoding overhead, guard-band overhead, and so on [29]. Furthermore, the control message overhead increases proportionally to the number of ONUs registered in a given EPON. Because more control frames will be transmitted per unit of time, reducing the cycle time further increases the observed control message overhead. To guarantee the delay for voice traffic in the access network at 1.5 ms, the cycle time should be about 1 ms [26,30].

$$Control\_Message\_Overhead = \frac{message\_size \times N_{ONU}}{cycle\_time \times EPON\_rate}, \tag{9}$$

where message_size is the size of GATE or REPORT message, $N_{ONU}$ is the number of ONUs, i.e., the number of messages sent in one cycle time, and the *EPON_rate* is the EPON transmission rate.

The proposed LSTM-DWBA can enhance the efficiency of upstream control overhead because the OLT begins sending the GATE message (transmission windows) to all ONUs in predictions cycle $I + 1$, without waiting for the REPORT message from ONUs, thus can reduce the control overhead, specifically upstream control overhead. As a result, predicting future bandwidth requirements is critical for ensuring the quality of service (QoS) of any tactile service, such as Tactile VR video streaming and teleoperation.

## 5. Performance Evaluation

To verify the effectiveness of the proposed system, we generate training data and conduct extensive simulations using the OPNET simulator. In the proposed approach we used 64 SD-ONU, and an SD-OLT with two wavelengths. The downstream/upstream channel rate between the SD-OLT and SD-ONU was dynamically assigned as 1 or 25 Gbps. The SD-OLT and SD-ONUs were uniformly distributed at 10–20 km apart, and the ONU buffer size was 10 Mb, which is shared by the priority queues for the class of services (CoS). To be more rigorous and realistic, bursty traffic is considered in our simulations. When a buffer is fully occupied and a packet with the higher priority arrives, the lower priority queue drops one or more packets to buffer a new packet. The maximum transmission cycles were 1.0 ms and 1.5 ms, and self-similarity and long-range dependence were used as the network traffic model for the TI, AF, and BE traffic, respectively, which generate high bursts of TI, AF, and BE traffic with a Hurst parameter of 0.7. The TI traffic distribution is Pareto and AF, BE packet traffic distribution is uniform and the packet size is between 64 and 1518 bytes, the model generated high-burst TI, AF, and BE traffic. The TI packet distribution is considered Pareto because it is control/steering and sensor packets. According to this bursty model, the resulting traffic is an aggregation of multiple streams each consisting of alternating Pareto-distributed ON and OFF periods. Pareto d. The probability density function of a generalized Pareto distribution can be expressed as follows [9]:

$$f(x) = \frac{\delta \mu^{\delta}}{x^{\delta+1}}, \tag{10}$$

where $\delta$ location parameter and $\mu$ is the shape parameter. The measurements on actual Ethernet traffic performed in estimate the Hurst parameter H for the Pareto distribution to be 0.7 for moderate traffic. The $\delta$ and $\mu$ of the Pareto distributions forming the ON and OFF periods are denoted as $\delta_{on}$ and $\delta_{off}$, respectively. Further, we follow the shape and location parameters according to ref. [9]. The traffic on the EF was based on a Poisson distribution with a fixed packet size (70 bytes) [31]. The DWBA computation time is 10 μs to grant the transmission timeslots on the SD-OLT and the guard time is set to 1 μs seconds to avoid overlapping of two sequential timeslots of different SD-ONUs. The summary of the simulation parameters can be seen in Table 1. To demonstrate the efficacy of high-priority traffic management, the three scenarios depicted in Table 2 were developed and analyzed with varying amounts of EF, TI, AF, and BE services. The traffic profiles are based on Sandvine (applications traffic forecast) report three geographic regions were simulated [32]. They are as follows Asia-Pacific (APAC); Europe-Middle East-Africa (EMEA); AMERICA. The traffic ratios of the LSTM-DWBA scheme were distributed as APAC S1-(10%, TI 6%, AF34%, BE 50%), EMEA S2-(10%, TI 7.5%, AF 42.5%, BE 40%), and AMERICA S3-(10%, TI 9%, AF 51%, BE 30%).

### 5.1. Dataset

To evaluate the feasibility of the proposed LSTM-DWBA scheme, we generate traffic data for the offline limited scheduling disciplines, which are the most extensively used legacy disciplines for predictive DWBA schemes, in this work. Our approach does not rely on a specific traffic arrival distribution (i.e., Poisson, Pareto, and Unform); rather, it is meant to estimate user demand regardless of the traffic arrival distribution. Simulations use Poisson and Self-Similar traffic, the most often used distributions in the literature. The data set includes the following nine features: EF Report, TI Report, AF Report, BE Report, EF Grant, TI Grant, AF Grant, BE Grant, and Cycle time. These were collected every 1.0 and 1.5 ms. This signifies that the observation was captured every 1 to 1.5 ms. Our dataset as a whole is made up of 200.000 data point samples that were gathered at all of the various network loads. This number of samples is substantial enough to construct robust LSTM models that generalize well. The training dataset will comprise 80% of the rows from the original data, while the validation dataset will account for the remaining 20%. The mean and standard deviation are applied to the dataset to standardize it. Based on the historical

past data of EF REPORT, TI REPORT, AF REPORT, and BE REPORT, the multi-Step model forecasts a range of future values, namely, TI GATE and AF GATE. Therefore, the LSTM multiple-step model forecasts a future sequence.

**Table 1.** Simulation parameters.

| Parameters | Value |
|---|---|
| Number of SD-ONUs | 64 |
| Number of wavelengths | 2 |
| Up/down link-rate | 1–25 Gbps |
| SD-OLT/SD-ONU distance | Uniform 10–20 km |
| Maximum transmission cycle time | 1 ms, 1.5 ms |
| Guard time | 1 μs |
| DWBA Computation time | 10 μs |
| ONU Buffer Size | 10 MB |
| EF Traffic distribution/Packet size | Poisson/70 bytes |
| TI Traffic Distribution | Pareto |
| AF, BE Traffic Distribution | Uniform |
| TI, AF, BE Packet Size | 64–1518 |

**Table 2.** Simulation scenario.

| Regions | Scenario | EF% | TI% | AF% | BE% |
|---|---|---|---|---|---|
| APAC | S1-DWBA/LSTM-DWBA (10%:40 (15%) 50 | 10% | 6% | 34% | 50% |
| EMEA | S2-DWBA/LSTM-DWBA (10%:50 (15%) 40 | 10% | 7.5% | 42.5% | 40% |
| AMERICA | S3-DWBA/LSTM-DWBA (10%:60 (15%) 30 | 10% | 9% | 51% | 30% |

The proposed LSTM-DWBA uses two LSTM layers with a dense layer of 5000 to forecast multi-steps. As the loss function, we use the mean squared error (MSE) between the projected and actual EF Grant, TI Grant, AF Grant, and BE Grant. Root mean squared propagation (RMSProp), Adam optimizer, gradient descent, and AdaGrad trained the models. LSTM hyperparameters are also used to improve the performance. Therefore, the optimal number of epochs to train our training dataset is 75 epochs. The TensorFlow and Keras backends with Scikit-learn (Sklearn) machine algorithms generate and train our LSTM regressors.

*5.2. Result and Analysis*

In this section, we analyze the performance of the proposed LSTM-DWBA (with prediction) under a limited scheme and compare it to the performance of a typically limited scheme (without prediction) for a number of traffic profiles. The effectiveness of the

proposed system is evaluated based on the mean packet delay, jitter, system throughput, and packet loss of the traffic.

### 5.2.1. Mean Packet Delay

Mean packet delay arises when packets arrive randomly at the ONU. Each packet should wait for the suitable upstream transmission time slot before it can be transmitted. This waiting period is known as the packet delay and is comprised of the polling delay, granting delay, and queuing delay [33]. Figure 6 depicts packet delay versus different traffic loads for the EF, TI, AF, and BE. The results demonstrate that with-prediction LSTM-DWBA improves the accuracy of traffic estimation, which in turn contributes to the reduction of delay compared to DWBA without prediction. Reduced packet delay is essential for providing QoS, especially for applications requiring a flawless user experience, such as Tactile applications. Using historical data, the LSTM-DWBA can reduce control message overheads by properly forecasting the EF, TI, AF, and BE REPORT messages next steps in advance. Moreover, EF, TI, and AF traffic delays are reduced by 1.0 ms compared to 1.5 ms, but only for the BE traffic delay was only reduced the 1.5 ms compared to 1.0 ms because of the reduced packet loss and improved bandwidth utilization.

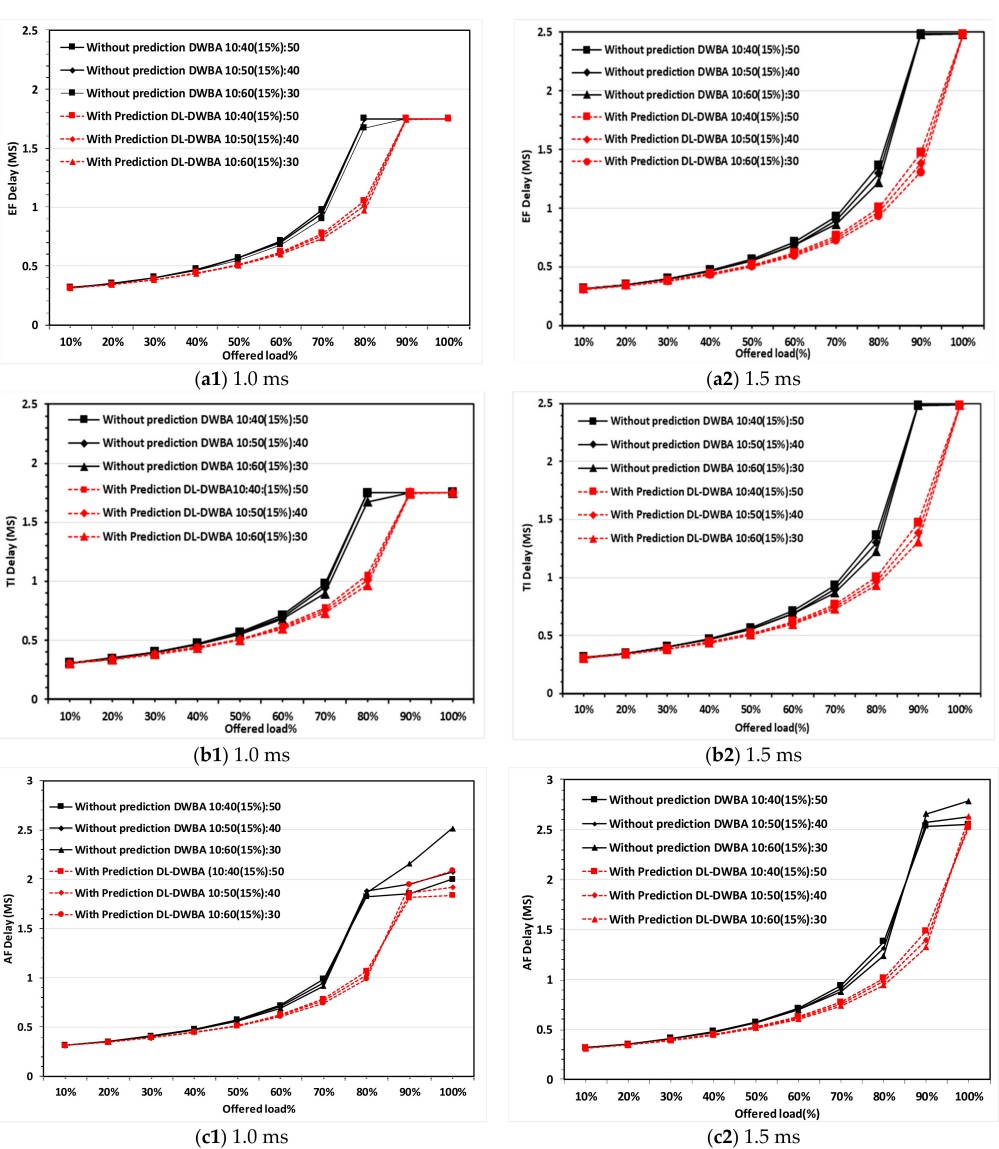

**Figure 6.** *Cont.*

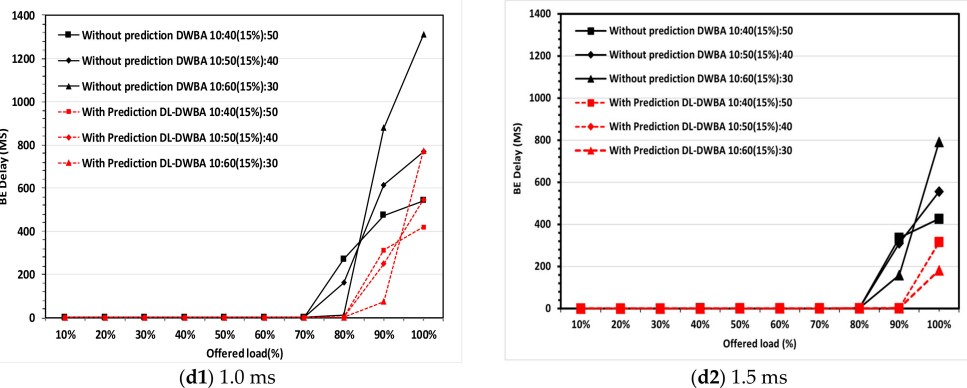

(**d1**) 1.0 ms　　　　　　　　　　　　　　(**d2**) 1.5 ms

**Figure 6.** Mean packet delay comparison in different traffic scenarios in 1.0 and 1.5 ms cycle time. (**a**) EF (**a1,a2**), (**b**) TI (**b1,b2**), (**c**) AF (**c1,c2**), and (**d**) BE (**d1,d2**).

### 5.2.2. TI Jitter

Jitter is crucial for the temporal performance of the network, as high latency renders interactive tactile applications useless, such as voice and two-way video teleconferencing. Figure 7 shows the mean TI jitter of with prediction LSTM-DWBA and without prediction LSTM-DWBA with different traffic loads. The TI jitter in LSTM-DWBA is improved for all scenarios compared to the without-prediction DWBA. The jitter of S1, S2, and S3 in LSTM-DWBA for TI traffic is less than 0.3%, indicating that TI packets are carried at almost identical intervals so that users can have continuous communications.

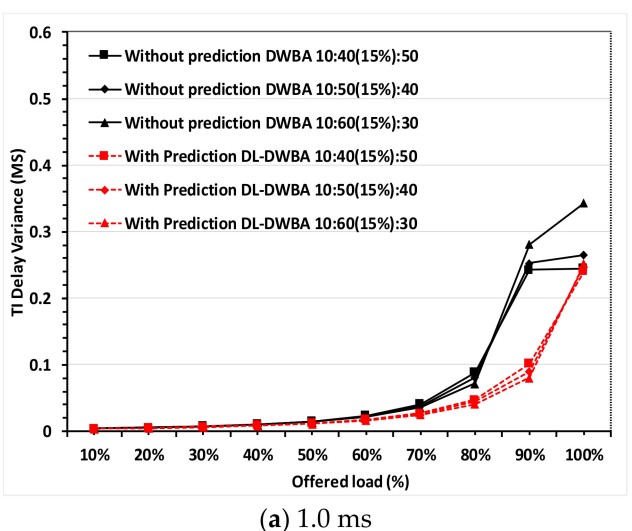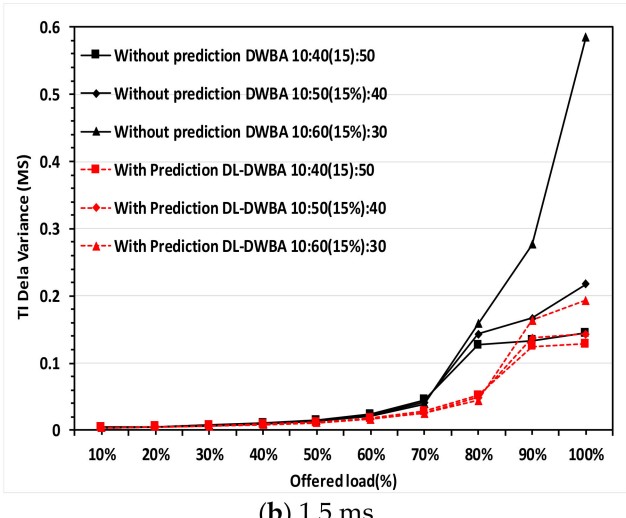

(**a**) 1.0 ms　　　　　　　　　　　　　　(**b**) 1.5 ms

**Figure 7.** TI Jitter performance.

### 5.2.3. System Throughput

Figure 8 shows the average system throughput with prediction LSTM-DWBA and without LSTM-DWBA with different traffic loads with 64 SD ONUs. The system throughput is defined as the sum of the data rates delivered to all network terminals. The system throughput in the proposed scheme is the sum of the communication throughput between ONUs and OLT and the local traffic throughput [34]. System throughputs are affected by cycle time, unused residual, and guard time. Because the LSTM-DWBA can foresee multiple steps, upstream overheads, such as guard time can be removed, resulting in increased upstream bandwidth efficiency. The LSTM-DWBA system's bandwidth consumption reaches 160% whereas without prediction DWBA reaches 141% in 1.0 ms, as shown in Figure 8a,b shows the 1.5 ms system bandwidth consumption reaches 179% whereas

without prediction DWBA reaches 160%. Moreover, the system throughput is increased by 1.5 ms compared to 1.0 ms.

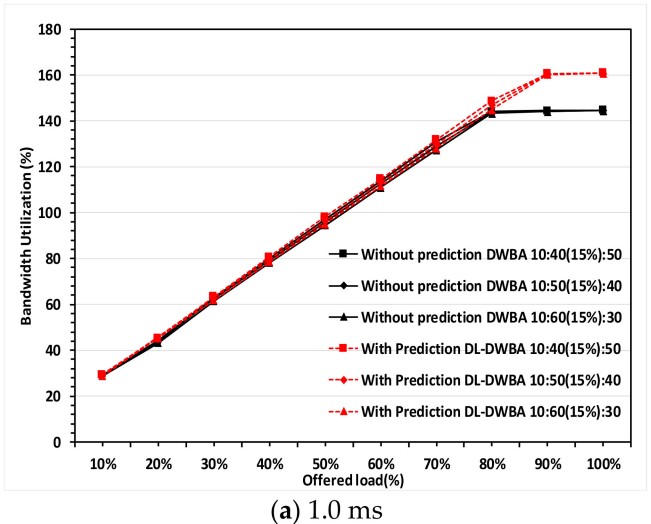
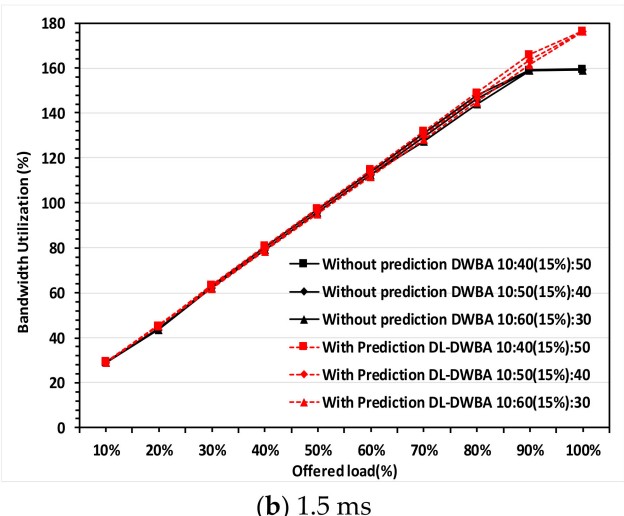

(**a**) 1.0 ms      (**b**) 1.5 ms

**Figure 8.** Overall system throughput.

### 5.2.4. Packet Loss

The packet loss is defined based on the traffic priority and buffer conditions and also cycle time. The EF, TI, and AF packet losses of the proposed with perdition LSTM-DWBA and without prediction DWBA were zero for all scenarios and traffic loads. BE packet loss was nil for both the with prediction LSTM-DWBA and the without prediction DWBA when the traffic load was less than 70%. To ensure the appropriate QoS performance in high-traffic circumstances, the low-priority BE packets are dropped if the buffer is full, as shown in Figure 9. In other words, when an ONU buffer is fully occupied and a packet with the higher priority arrives, the lower priority queue drops one or more packets to buffer a new packet. In addition, scenarios with a cycle time of 1.5 ms had a reduced packet loss percentage because ONUs had more time to broadcast their buffered packets. Moreover, Figure 9a 1.0 ms of all packet loss is higher than 1.5 ms because BE traffic has the lowest priority in our DBA. Our traffic priority is EF, TI, AF, and BE. The DBA first satisfied the higher-priority traffic (EF, TI, and AF); after that, it allocated the bandwidth to BE so it could take the longer time slots. Our proposed prediction DWBA packet transmission time slot was shorter BE traffic in 1.0 ms, and a packet loss of nearly 5% was achieved. In addition, the mean packet delay (Figure 6d1) increased by nearly 800 ms. At the same time, the packet loss in 1.5 ms is reduced below 1.2%, and the mean packet delay is also reduced below 400 ms in Figure 6d2. Therefore, our analysis of packet loss shows that packet loss is reduced and bandwidth utilization is increased as the cycle time increases. In 1.5 ms, our proposed DWBA reduces packet drop to less than 1.2%. In the event of heavy traffic loads, the BE packet losses of the proposed LSTM-DWBA and DWBA were identical across all scenarios and cycle periods, indicating that the proposed architecture will not affect packet loss. In the event of heavy traffic loads, the BE packet losses of the proposed LSTM-DWBA and DWBA were identical across all scenarios and cycle periods, indicating that the proposed architecture will not affect packet loss.

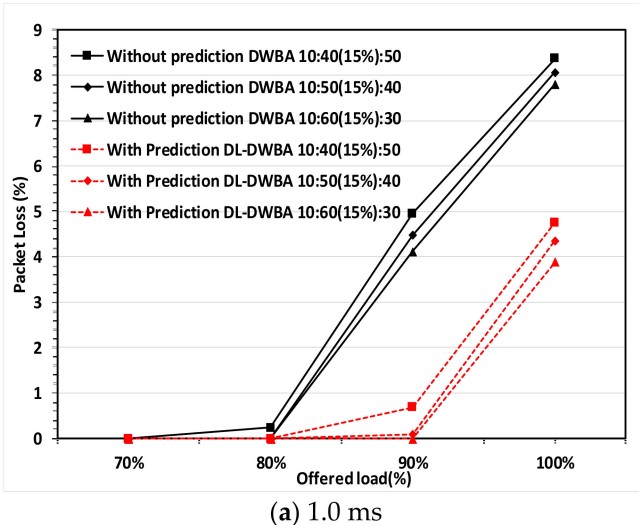
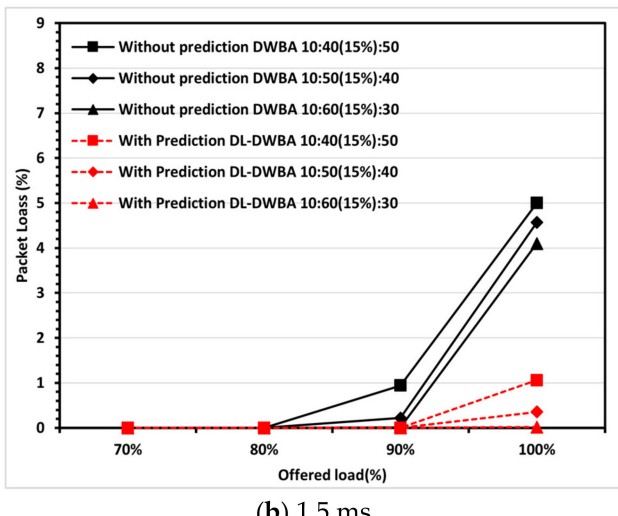

(**a**) 1.0 ms            (**b**) 1.5 ms

**Figure 9.** Overall system packet loss.

## 6. Conclusions

In this paper, we proposed an LSTM-DWBA scheme for SD-NG-EPONs-based tactile applications that use LSTM-RNN to predict future SD-ONU bandwidth demands based on past SD-ONU demands. Based on historical data, LSTM-DWBA can forecast future EF, TI, AF, and BE bandwidth requirements. The proposed LSTM model achieved high accuracy with a negligible MSE. The prediction-based LSTM-DWBA scheme reduced the control message overhead and improved the QoS services of tactile applications. Moreover, the results are demonstrated at 1.0 ms and 1.5 ms cycle times. Future work can use more ML models to improve the network performance.

**Author Contributions:** Conceptualization and methodology, E.G. and A.T.L., formal analysis S.W.T., validation and editing M.S.A.-R. and M.N.A.S., supervision and funding, I.-S.H. All authors have read and agreed to the published version of the manuscript.

**Funding:** This work was funded by the National Science Council under grants MOST 109-2221-E-155-029-MY2.

**Institutional Review Board Statement:** Not applicable.

**Informed Consent Statement:** Not applicable.

**Data Availability Statement:** Not applicable.

**Conflicts of Interest:** The authors declare no conflict of interest.

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
