# Peer review of "LSTM-Based DWBA Prediction for Tactile Applications in Optical Access Network"

_photonics, doi:10.3390/photonics10010037_

Round 1

Reviewer 1 Report

The following points should be addressed:

-The innovative contributions of the manuscript should be made clearer;

-The state of the art should be extended because many contributions investigated the performance of LSTM-based neural networks [1,2];

-LSTM is very complex and time consuming solution and its use should be better justified;  

-The performance should be evaluated in realistic traffic scenario;

-Effectiveness benchmark solutions should be considered for the comparison; the authors compare onlywith a solution in which traffic prediction is not performed;

[1] V. Eramo, F.G. Lavacca, T. Catena, P. J. Perez Salazar: Application of a Long Short Term Memory neural predictor with asymmetric loss function for the resource allocation in NFV network architectures, Computer Networks (Elsevier) vol. 193, July 2021, pp. 108104-108116

[2] V. Eramo, T. Catena: Application of an Innovative Convolutional/LSTM Neural Network for Computing Resource Allocation in NFV Network Architectures, IEEE Transactions on Network and Service Management, vol. 19, September 2022, pp. 2929 - 2943

Author Response

FYI!!

Reviewer 2 Report

The paper investigates the LSTM-Based DWBA prediction for tactile applications in optical access network. I think the paper is interesting to the readers. The comments are as following.

1. The affiliation of some authors are lost ex. the university name.

2. The number of conclusion should modify, it should be number "6".

3. The peak loss should give some error analysis.

4. Some data of figures are too closely, it should give a table to present the data, ex. Figures 7 and 8.

5. Please give explanations for the parameters of the Table 1 on section Performance Evaluation.

Author Response

FYI!!

Round 2

Reviewer 1 Report

All my comments have been discussed and addressed.